# Parameterized Pseudo-Differential Operators for Graph Convolutional Neural Networks

## Abstract

We present a novel graph convolutional layer that is fast, conceptually simple, and provides high accuracy with reduced overfitting. Based on pseudo-differential operators, our layer operates on graphs with relative position information available for each pair of connected nodes. We evaluate our method on a variety of supervised learning tasks, including superpixel image classification using the MNIST, CIFAR10, and CIFAR100 superpixel datasets, node correspondence using the FAUST dataset, and shape classification using the ModelNet10 dataset. The new layer outperforms multiple recent architectures on superpixel image classification tasks using the MNIST and CIFAR100 superpixel datasets and performs comparably with recent results on the CIFAR10 superpixel dataset. We measure test accuracy without bias to the test set by selecting the model with the best *training* accuracy. The new layer achieves a test error rate of 0.80% on the MNIST superpixel dataset, beating the closest reported rate of 0.95% by a factor of more than 15%. After dropping roughly 70% of the edge connections from the input by performing a Delaunay triangulation, our model still achieves a competitive error rate of 1.04%.

## 1 Introduction

Convolutional neural networks have performed incredibly well on tasks such as image classification, segmentation, and object detection (Khan et al., 2020). While there have been diverse architectural design innovations leading to improved accuracies across these tasks, all of these tasks share the common property that they operate on structured Euclidean domain inputs. A growing body of research on how to transfer these successes into non-Euclidean domains, such as manifolds and graphs, has followed.

We focus on unstructured graphs which represent discretizations of an underlying metric space. These data types are ubiquitous in computational physics, faceted surface meshes, and (with superpixel conversion) images. Previous efforts to extend CNNs to this type of data have involved parameterized function approximations on localized neighborhoods, such as MoNet (Monti et al., 2017) and SplineCNN (Fey et al., 2017). These function approximations (Gaussian mixture models in the case of MoNet and B-spline kernels in the case of SpineCNN) are complex and relatively expensive to calculate compared to CNN kernels.

Inspired by earlier work in shape correspondence (Boscaini et al., 2016), image segmentation on the unit sphere (Jiang et al., 2019), and low-dimensional embeddings of computational physics data (Tencer & Potter, 2020) we seek to utilize parameterized differential operators (PDOs) to construct convolution kernels. In contrast to MoNet and SpineCNN, parameterized differential operators are cheap to compute and involve only elementary operations. Boscaini et al. (2016) used anisotropic diffusion kernels while Jiang et al. (2019) included gradient operators in addition to an isotropic diffusion operator. Tencer & Potter (2020) performed an ablation study of the differential operators used and demonstrated that the including the gradient operators is broadly beneficial, but that little is gained by including additional terms.

Prior work (Jiang et al., 2019; Tencer & Potter, 2020) has used differential operators precomputed for specific meshes. This approach has two drawbacks: (1) precomputing operators is not practical

for datasets for which the connectivity graph varies between sample points, and (2) differential operators place restrictions on graph connectivity. Differential operators defined for mesh topologies rely on element connectivity information which is unavailable for more general graphs. Superpixel image datasets highlight both of these deficiencies. In contrast to these prior works, we do not precompute any operators and we do not directly use differential operators. Instead, we formulate *pseudo-differential* operators which are cheaply computed at run-time for a more general class of graphs.

While our approach only applies to graphs with relative position information for each node, the set of graphs with the required positional information is large, encompassing nearly all physical systems as well as a significant number of other graphs, such as graph representations derived from image data.

Since our method relies on computing approximate spatial derivatives of nodal features, it is also important that these nodal values represent a meaningfully changing field. This criteria is not necessarily met for the node correspondence task on the FAUST dataset or the shape classification task on the ModelNet10 dataset and a corresponding decrease in performance is observed. In contrast, nodal features are critical to superpixel image classification tasks and our method is observed to perform well for these datasets.

Superpixel representations are popular for a wide range of tasks, particularly tasks in which large data quantities make the direct application of CNNs to the raw data impractical, such as hyperspectral imaging (Hong et al., 2020) and medical diagnostics (Roth et al., 2015). For these applications, the superpixel representation serves as a sort of context aware lossy compression. Knyazev et al. (2019) compared GCNs applied to superpixel images to CNNs applied to low resolution images with approximately the same information content. In those cases, the graph methods not only held their own, but pulled ahead compared to the CNN performance. While those datasets were not at all pushing the limitations of image size, they indicate a possibility that superpixel methods might handle high resolution image data more efficiently and the value in developing methods that perform well on superpixel datasets.

Our method is especially well-suited for analyzing superpixel image representations in addition to being applicable to the datasets used by Jiang et al. (2019) and Tencer & Potter (2020) to demonstrate their PDO-based approaches. For regular meshes, such as the icosahedral spherical mesh used by Jiang et al. (2019), our pseudo-differential operators closely approximate the differential operators used in those works.

### 1.1 OUR CONTRIBUTIONS

We created a novel layer architecture inspired by PDOs.

- We improve upon the static matrix approach of Tencer & Potter (2020) with a dynamic method that enables support for variable graph forms and eliminates the need to precompute matrices.
- Our method utilizes pseudo-differential operators in contrast to the differential operators used in prior works. Pseudo-differential operators are cheap to compute and are applicable to a broader class of graphs than differential operators.
- Our novel mixing layer is conceptually simple and easy to code (integrating painlessly with existing graph libraries). (section 3.1)
- The new approach remains accurate for both sparsely and densely connected graphs, including state-of-the-art results for the MNIST superpixel 75 dataset both with and without reduced edge connection input data. (section 4.1)
- The new approach is faster than common approaches for equivalent numbers of features owing to the simpler mathematical functions involved. (section 4.1.2)

## 2 PROBLEM

Many real world datasets may be treated as attributed graphs with positional information provided for the nodes or relative positional information provided for the edges. In physics and engineer-

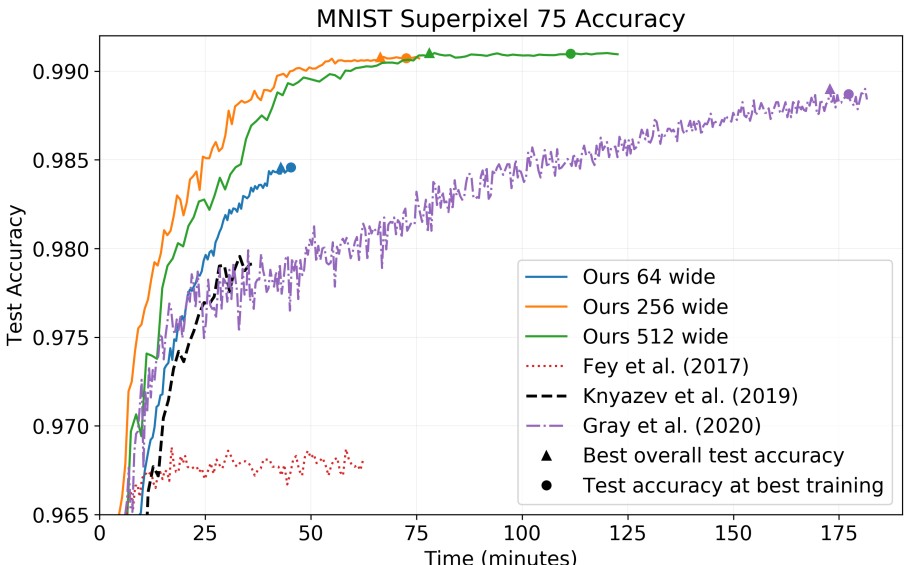

Figure 1: Accuracy compared to training time. While some of the other methods perform slightly better than our 7 layer deep model in time per epoch, we achieve higher accuracies per training minute (in addition to achieving a higher overall accuracy). Of note, using a wider network is not a penalty in accuracy vs training time. Plots show the mean test accuracy/time over 10 runs — except Gray et al. (2020), which shows the median results from 10 runs due to their higher variability.

ing, high-fidelity simulation tools represent continuous spatial functions using unstructured spatial discretizations. In computer vision, triangle surface meshes and super pixel image representations are common. Tencer & Potter (2020) have recently used convolutional autoencoder architectures with PDOs to perform manifold learning on datasets from computational physics. Their approach required precomputing matrices corresponding to various differential operators on a sequence of progressively coarser meshes. We sought to generalize their approach to support heterogeneous graphs without explicitly defined gradient operators and to avoid the need to precompute matrices or coarsened spatial discretizations.

Our ideal solution, would be fast, simple, scale well, make efficient use of information, and operate effectively on sparsely connected graphs. While there are a number of existing approaches that will work for arbitrary graph information, all have some form of limitation that makes them ill suited for our use cases. Whether ignoring graph connectivity information (Gray et al., 2020), non-existent use of position information (Kipf & Welling, 2016; Berg et al., 2017), or complex formulations (Fey et al., 2017; Monti et al., 2017), each has some weakness that made its use less than ideal.

## 3 METHOD

Our layer works by calculating several quantities that are analogous to the sets of differential operators used in Tencer & Potter (2020). For each node of a layer, each input channel has multiple values calculated (4 items for 2-dimensional and 5 for 3-dimensional graphs):

- Value of the node in the prior layer (identity)

- Average values at the 1-ring neighbors of the node

- Average gradient components (i.e. $\frac{\partial}{\partial x}, \frac{\partial}{\partial y}$) of value along each connected edge weighted by inverse edge length

These items are then mixed by a neural network yielding the desired number of output channels.

### 3.1 NOVEL LAYER

In contrast to previous methods utilizing PDOs, the method presented here generalizes to heterogeneous datasets, in which the number of nodes, the connectivity, and positions vary across samples. Additionally, by relaxing the definitions of the gradient operators, our implementation is applicable to overconnected graphs (such as superpixel image representations) rather than only meshes suitable for common PDE solution methods (finite element, finite volume, etc.). A consequence of this generalization is slightly more computational overhead from dynamically generating the required operators on the fly rather than precomputing them offline. However, as seen in Figure 1, this overhead does not result in slower training time relative to other methods.

The key components of Tencer & Potter (2020)'s approach were the sparse matrix mixing, pooling, and interpolation. In our implementation of the mixing layer, we used pytorch-geometric's `MessagePassing` base class Fey & Lenssen (2019). For pooling layers, we evaluated two of pytorch-geometric's clustering method implementations: `voxel_grid` and `graclus` (Dhillon et al., 2007; Simonovsky & Komodakis, 2017). Their `voxel_grid` implementation gave the best accuracy on the MNIST superpixel dataset.

The `MessagePassing` class operates by calculating new values for each node $i$ on layer $^{(k)}$ with information from their prior node values $x_i^{(k-1)}$, prior values at connected nodes $x_j^{(k-1)}$, and/or the edge attributes $e_{i,j}$.

$$\mathbf{x}_i^{(k)} = \gamma^{(k)} \left( \mathbf{x}_i^{(k-1)}, \square_{j \in \mathcal{N}(i)} \phi^{(k)} \left( \mathbf{x}_i^{(k-1)}, \mathbf{x}_j^{(k-1)}, \mathbf{e}_{i,j} \right) \right) \tag{1}$$

$\square_{j \in \mathcal{N}(i)}$ is an aggregation method (such as mean) that aggregates $\phi^{(k)}$ for each node $j$ connected to node $i$ in layer $^{(k-1)}$ into a specific number of values independent of the number of connections. $\phi^{(k)}$ and $\gamma^{(k)}$ are arbitrary functions.

Convenient and physically motivated choices for $\phi^{(k)}$ are derived from mesh differential operators, e.g. $I$, $\nabla$, or $\Delta$. The identity operator $I$ simply passes the value forward. For the others, let $f : \Omega \to \mathbb{R}$ be a smooth scalar function for which only the values $f_1, \ldots, f_n$ at the nodes are known and $f_i$ corresponds to the value of $f$ at node $i$. The Laplacian operator $\Delta$ may be expressed as the difference between $f_i$ and the average value of $f_j, j \in \mathcal{N}(i)$. For triangle meshes, the nodal gradient operator $\nabla$ is often expressed as the average gradient in the adjacent facets which is equivalent to a weighted sum over the connected edge gradients

$$\nabla f_i \approx \sum_{v_j \in \mathcal{N}(i)} w_{i,j} \nabla f(e_{i,j}). \tag{2}$$

with $\nabla f(e_{i,j}) = \frac{f_i - f_j}{r_{i,j}} \hat{e}_{i,j}$. We denote the Euclidean distance between the two nodes as $r_{i,j}$, and the unit vector oriented along the edge $e_{i,j}$ as $\hat{e}_{i,j}$. For a 2D mesh without intersecting edges $w_{i,j} = \frac{A_{i,j}}{\sum_{j \in \mathcal{N}(i)} A_{i,j}}$, where $A_{i,j}$ is the total area of the 2 facets connected to $e_{i,j}$ (Mancinelli et al., 2019). For an overconnected graph (like MNIST superpixel), these facet areas are not defined.

If we weight the contributions of each edge equally, our choice for $\phi^{(k)}$ becomes

$$\frac{\left( \mathbf{x}_i^{(k-1)} - \mathbf{x}_j^{(k-1)} \right) r_{x,i,j}}{r_{i,j}^2}, \frac{\left( \mathbf{x}_i^{(k-1)} - \mathbf{x}_j^{(k-1)} \right) r_{y,i,j}}{r_{i,j}^2}, \mathbf{x}_j^{(k-1)}, \tag{3}$$

where $r_{x,i,j}$ and $r_{y,i,j}$ are the differences in positions of nodes $i$ and $j$ in the x- and y-dimensions, respectively. We can easily extend these to 3 dimensions (which was required for the FAUST dataset) by adding a z term, in which case, the parameter cost per channel goes up by only a factor of 25%. $\phi^{(k)}$ returns a stack of these values for every input channel. The first 2 terms of equation 3 are x- and y-components of the gradient. The $3^{rd}$ term is the average of the neighboring nodes, which, when combined with the identity term $x_i^{(k-1)}$ (which is concatenated after aggregation) results in the Laplacian. Given that the next step in the layer is to mix these components via a neural network (our $\gamma^{(k)}$ function), the Laplacian can be reconstructed by blending these terms if desirable.

Note that none of these values require complex calculations, which contributes to the layer's superior computational performance (section 4.1.2). We hypothesize that by limiting the representational

space for each node such that it knows only about itself and the local gradient the network is forced to find simpler representations that are more likely to generalize.

# 4 EXPERIMENTS

We tested against 3 variants of MNIST superpixel 75, both CIFAR10 and CIFAR100 as superpixels, FAUST, and ModelNet10 (Monti et al., 2017; Krizhevsky et al., 2009; Bogo et al., 2014; Wu et al., 2015). For the remainder of this paper MNIST superpixel 75 will be referred to as MNIST. In the absence of validation sets, we selected test results based on the best training accuracy in order to avoid biasing our model selection to the test set. For our models this is often comparable, but is not necessarily the overall best test accuracy (figure 1). While we considered using k-fold cross validation, we found that our method was effective in picking out generally applicable models without biasing to the test set. It was both conceptually simpler and easier to implement as well.

SplineCNN was taken from examples in pytorch-geometric (Fey & Lenssen, 2019) and utilized early stopping to avoid overfitting. For fair comparison, their best overall test accuracy was chosen. For MNIST and CIFAR superpixel, we applied Gray et al. (2020) and Knyazev et al. (2019). We reported results using the test accuracies for the best training accuracy and note any deviations.

The graph attention convolution layer (GATConv) (Velikovi et al., 2018) was taken from pytorch-geometric's implementation. GATConv networks have been recently applied to the MNIST and CIFAR-10 superpixel benchmarks (Avelar et al., 2020). For both MNIST and CIFAR-10, the published GATConv results are inferior to the other comparison models. We tested a variety of head and channel hyperparameters against a version of the SplineCNN model with the spline convolutions replaced with GATConv. We replicated their results for MNIST.

There is some variation in results across our model initializations. The trend of high test performance following high training set accuracy was robust. This allowed a bias-free way of selecting performant models from multiple runs of the same hyperparameters.

For all training runs, our models had training accuracies on par with test accuracies. Even MNIST had training results at or below test accuracy with appropriate levels of edge dropout (section 4.1.1).

All models were trained with Adam optimization using a learning rate of 0.0002 and cross entropy loss for our architecture and the published options for comparison models unless otherwise noted. Learning rates were reduced by a factor of 10 on a plateau of 5 epochs without improving training loss up to a limit of $1/1000^{th}$ the original learning rate.

## 4.1 MNIST SUPERPIXEL 75

We tested against 3 variants of the MNIST dataset (Monti et al., 2017). The first variant used all of the edges present in the original graph which we call the *raw* dataset. The second, we use a variant of the *hierarchical* set used in Knyazev et al. (2019) with sets of approximately 75, 21, and 7 superpixels forming the feature hierarchy for each sample. The number of edges for each node was limited to its 32 closest by Euclidean distance as we are not focused on optimizing performance for minimal input sizes. Lastly, our *pruned* version was obtained by discarding all of the edge data and applying Delaunay triangulation to the nodes, which reduced the number of edge connections by around 70% on average.

Our architecture remained fairly consistent throughout testing. The model was comprised of between 3 and 7 layers of downsampling modules followed by 2 fully connected layers with an exponential linear unit between the two. In the end, we selected 7 of our layers for all main results because it provided the best accuracy. Results for shallower networks are in appendix A. Each downsampling module consisted of our introduced graph convolutional layer followed by a voxel grid pooling operation outputting a reduced set of nodes with a selectable count of features. Pooling is done with voxel size halved at each step. Initial voxel size is set such that after the 7 downsampling operations, a $3 \times 3$ set of nodes is present prior to flattening for the fully connected layers.

No normalization methods were included in the downsampling layers (e.g. batch, dropout, edge dropout, etc.). Standard dropout of 0.5 was applied prior to all the fully connected layers. When used, we applied edge dropout to the data prior to model input.

Table 1: MNIST superpixel 75 image classification results. Our model and Fey et al. (2017) were trained for 100 epochs on the MNIST dataset. Knyazev et al. (2019) was trained for 30 epochs and Gray et al. (2020) for 400 epochs, keeping to their implementations. To avoid biasing to the test set, we selected the models by best training accuracy for a given set of hyperparameters — with the exception of Fey et al. (2017) for which we chose the overall best for each particular run. Average and standard deviation values taken across 32 runs for the raw and hierarchical variants, and against 10 for the pruned variant. Given that Knyazev et al. (2019) was highly dependent on the extra information from their hierarchical representation, we did not run their code against the pruned version. Velikovi et al. (2018) produced a graph attention convolutional layer but when applied to MNIST it performed poorly (and was worse without embedding positions into the input vector). Fey et al. (2017) overfit badly when using the hierarchical dataset. Gray et al. (2020) reports an accuracy of 99.05% on the raw superpixel dataset, which we were only able to replicate by taking the overall maximum test accuracy out of 5 runs, rather than the average.

|  | Ours | Velikovi | Fey | Knyazev | Gray |
|---|---|---|---|---|---|
| Raw superpixel | **99.12**±0.06 | 94.70±0.23 | 97.05±0.22 | 97.11±0.22 | 98.81±0.04 |
|  | **99.20**[1] | 95.01[2] | 97.45[2] | 97.44[1] | 98.86[1] |
| Hierarchical | **99.04**±0.05 |  |  | 98.29±0.21 | 98.26±0.09 |
|  | **99.12**[1] |  | 69.75[2] | 98.55[1] | 98.38[1] |
| Pruned | 98.78±0.13 |  | 96.95±0.11 |  | **98.82**±0.07 |
|  | **98.96**[1] |  | 97.19[2] |  | 98.89[1] |

Our architecture with the best performance on all the MNIST superpixel variants used 7 downsampling layers starting with 128 features, doubling each layer, until a maximum width of 512 features. Edges were dropped out on the input to prevent overfitting to the training set (section 4.1.1).

Our architecture achieved a state-of-the-art test error of **0.80%** against the raw MNIST dataset using our best training accuracy as a selector among 32 runs. The average results for ours and several competitors are shown in Table 1. Our architecture achieved these results using an edge dropout rate of 0.45 and a learning rate of 0.002, although runs with edge dropout rates between 0.3 and 0.55 were competitive or beating state-of-the-art in a large number of the training runs (section 4.1.1). This compares to the prior best reported value of 0.95% from Gray et al. (2020). We were able to reproduce their result but only by selecting the best overall test accuracy out of 5 runs. For the pruned variant, when choosing the model based on best training accuracy, our model's test error outperforms results from Gray et al. (2020) with an error rate of 1.04%.

For the hierarchical variant, we note a slight drop in accuracy relative to the raw dataset. The hierarchical dataset adds additional nodes for various scales of abstraction (parent, grandparent, etc) with each level being progressively coarser. Knyazev et al. (2019) explicitly treats each of these levels differently, recognizing that the edge connection is between a child-parent, siblings, etc. as part of its multigraph convolution approach. None of the other methods differentiate between these edge relationships. In our case, we hypothesize that the extra information can cause issues with the Laplacian term as it does not have a distance weighting. The extra nodes have reduced quality information which we believe can act as a confuser to the networks when the connection type is ignored. For Fey et al. (2017), the extra information causes extreme overfitting to the training set.

### 4.1.1 EDGE DROPOUT

To understand how reducing the edge connection through pruning impacted accuracy, we trained against the raw MNIST dataset with varied rates of edge dropout applied to the input data. As shown in Figure 2a, our method beats the comparison models on the raw dataset for a range of rates.

At dropout rates above 0.6, orphaned nodes become much more likely and are a significant portion of the nodes by 0.8 dropout. These orphaned nodes negatively impact our performance as the only data left to pass forward at each orphaned node is the identity function. Given that position information is

---

[1] Test accuracy selected from best training epoch
[2] Best overall test accuracy selected

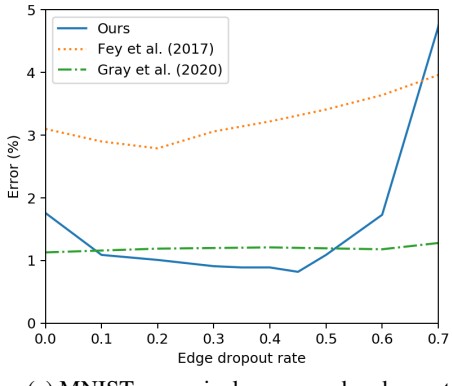

(a) MNIST superpixel error vs edge dropout

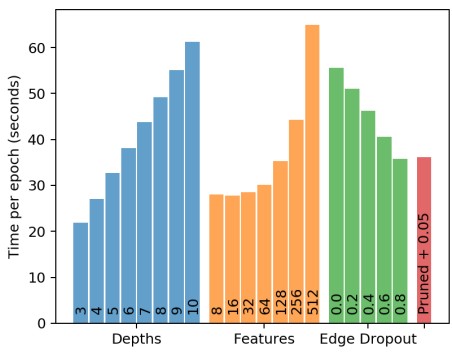

(b) Hyperparameters effect on performance

Figure 2: (a) MNIST superpixel error rates as graph edges are dropped on input. Our model performs significantly better for a range of edge dropout rates. Gray et al. (2020) use a technique which ignores the input edges which leads to it being unaffected by edge dropout. Fey et al. (2017) sees some benefit from low edge dropout rates (probably from reduced overfitting). (b) Effect of hyperparameters on performance of our architecture. We modify a default set of parameters — 7 layers deep, 256 initial & max features, no pruning, and 0.5 edge dropout. Each was run for 5 epochs on a nVidia V100 and average times per epoch are shown.

required by our method, an appropriate and effective edge network can be acquired using Delaunay triangulation for 2D graphs. Pruning the MNIST graphs in this way results in approximately a 70% reduction in edges. Using the pruned MNIST dataset, our code performs competitively with prior reported state-of-the-art test accuracy (against models trained and evaluated using the raw superpixel dataset). When comparing test accuracy from the model selected by training accuracy, we outperform other models trained and tested using the raw dataset. We achieved an error rate of **1.04%** using a 0.1 edge dropout applied on the pruned input graph as shown in Table 1. Because of this, the performance degradation for overly sparse graphs (high edge dropout rates) should not be a practical limitation for our method.

Moderate levels of input edge dropout seems to provide a data augmentation effect for the raw MNIST dataset, leading to greater test accuracy and less overfitting (our train accuracies remain comparable to test when sufficient edge dropout is used). In all cases, our architectures performed better with some level of input edge dropout than with the original graph. In addition, reduced edge counts have a beneficial impact on training times (section 4.1.2).

Gray et al. (2020) uses a novel method which ignores incoming edge information. Because of this, its accuracy is the same for all levels of edge dropout and pruning (within their normal variance). While this eliminates any accuracy penalty for overly sparse graphs, it also eliminates the data augmentation and performance benefits of input edge dropout.

### 4.1.2 PERFORMANCE

Our network is faster on a per epoch of training time per layer basis. After adding layers to create a deeper, wider, and more performant network, it remains competitive in training time and superior when looking at test accuracy achieved per training time as shown in Figure 1. Surprisingly, adding extra width to the network did not worsen convergence time until a slight dip is observed going from 256 to 512 features wide. While each epoch takes longer, it converges faster.

We also show the impact of various hyperparameters on our network performance in Figure 2b . Of note, our pruned dataset running with 0.05 edge dropout has similar performance to the 0.8 edge dropout without pruning. The pruned dataset has around 70–75% fewer edge connections than in the original which accounts for the increase in speed.

Table 2: (a) CIFAR100 superpixel image classification test accuracy (%). We show the average and maximum test accuracies for the best training epoch over 32 runs. (b) FAUST node correspondence results. All results (except ours) as reported by authors. We did not perform an extensive hyperparameter search and our results are mostly to show that the method is generally applicable.

| (a) CIFAR100 accuracy | | | (b) Faust node correspondence | |
|---|---|---|---|---|
| Model | Superpixel | Hierarchical | Model | Reported Accuracy |
| Ours | **40.39**±0.27 **40.71**[3] | **41.16**±0.32 **41.41**[3] | Boscaini et al. (2016) | 62.4% |
| | | | Monti et al. (2017) | 73.8% |
| Knyazev | 30.46±1.07 31.41[3] | 32.34±0.84 33.54[3] | Sun et al. (2018) | 96.9% |
| | | | Verma et al. (2017) | 98.7% |
| | | | **Ours** | 99.20% |
| Gray | 34.28±0.44 34.88[3] | 34.57±0.45 35.19[3] | Fey et al. (2017) | 99.20% |
| | | | Gong et al. (2019) | 99.8% |
| | | | Haan et al. (2020) | **99.89%** |

## 4.2 CIFAR100 SUPERPIXELS

We also tested against CIFAR10 and CIFAR100 (Krizhevsky et al., 2009) converted to superpixels. The CIFAR100 results are shown in table 2a while the CIFAR10 results are available in appendix A . We achieved a best accuracy of 40.71% on the raw variant and 41.41% on the hierarchical variant.

The CIFAR100 superpixel dataset represents a significantly more challenging learning task than either standard (non-superpixel) CIFAR100 or MNIST. CIFAR100 is a set of $32 \times 32$ color images across 100 categories. Each category has 600 images, broken into 500 for training and 100 for test. The superpixel and hierarchical variants are generated by applying a SLIC transformations as described in Knyazev et al. (2019). Each image was constructed with approximately 150 superpixels with node edges restricted to only its 32 nearest neighbors by Euclidean distance. The nodes were constructed from levels of approximately 150, 75, 21, and 7 superpixels in the same manner as the hierarchical MNIST dataset (section 4.1). The CIFAR10 and CIFAR100 superpixel experiments used the same architecture as used for the MNIST experiments, with the exception of additional input channels for color superpixels and output channels for the larger number of classes in CIFAR100.

## 4.3 FAUST

We also tested our method on a shape correspondence task using the FAUST (Bogo et al., 2014) dataset, which contains 100 meshes with $6,890$ nodes each depicting 10 scanned human bodies in 10 different poses. Each node corresponds to a particular part of each body and the task is to identify which node corresponds to what body part. We used the standard 80/20 training/test split.

A modified version of the architecture used in MNIST experiments was used with 3-D gradients, 8 layers, 16 initial features doubling each layer to a maximum of 128, and ending with 2 fully connected layers (dropout applied before each). Scaled exponential linear units were used as the activation function between each layer. No flattening or pooling operations were used. The final output is a softmax with $6,890$ channels per node. Training used a batch size of 4, dropout of 0.3, learning rate of 0.01, and cross entropy loss. For this task, input edge dropout caused a significant performance drop and was not used. Although no extensive hyperparameter tuning was performed, we achieved an accuracy performance comparable with recent results of 99.20% as shown in table 2b.

## 4.4 MODELNET

We also test our method on the shape classification task using the ModelNet10 (Wu et al., 2015) dataset, which contains $4,899$ 3D CAD models of objects from 10 categories. Our method performs worse on this task, achieving a middling classification accuracy of 88.6% on the test set (table 5). We attribute this poor performance to two factors. First, the ModelNet10 dataset does not include

---

[3] Test accuracy selected from best training epoch

nodal features. We attempted using the node positions and normal vectors as input features, but suspect that these were insufficient. Second, and perhaps more significantly, many of the meshes in the dataset contain large local variations in edge length. Consequently, our approximate derivatives computed via pseudo-differential operators devalue contributions from distant nodes creating information bottlenecks in the network. This is, perhaps, not an insurmountable obstacle, but resolving these issues is beyond the scope of this work.

## 5 RELATED WORK

Early approaches showed that CNNs could be used on non-Euclidean domains by introducing new intrinsic convolutional methods (Masci et al., 2015; Boscaini et al., 2016) that operate on input manifolds. More recent approaches like Monti et al. (2017) are capable of performing well on both manifolds and general graphs as input. Monti achieves this by creating pseudo-coordinates for either the vertices of a graph or points on a manifold, and then learns a kernel in that space.

Graph Convolutional Neural networks form the basis for other approaches that have shown great results (Zhang et al., 2019). The literature has been split among methods based on spectral graph theory (Bruna et al., 2013; Henaff et al., 2015; Defferrard et al., 2016; Kipf & Welling, 2016; Levie et al., 2017; Monti et al., 2017; Wang et al., 2018; Tencer & Potter, 2020) and methods that operate with spatial filters (Micheli, 2009; Atwood & Towsley, 2015; Niepert et al., 2016; Fey et al., 2017; Gilmer et al., 2017; Gray et al., 2020; Velikovi et al., 2018). Our method falls into the latter category of spatial approaches.

## 6 CONCLUSION

We introduced a simple graph convolutional layer that outperformed every published result we are aware of for superpixel image classification on the MNIST and CIFAR100 superpixel datasets. We demonstrated faster performance and reduced overfitting tendencies with the MNIST dataset as well. In addition, we tested our new layer on the CIFAR10 superpixel image classification, FAUST shape correspondence, and ModelNet10 shape classification datasets. On CIFAR10 superpixel and FAUST we achieved accuracies comparable to recent results without significant hyperparameter tuning. However, with ModelNet10 our layer only demonstrated moderate performance.

Input edge dropout often provided positive impacts on our accuracy except at extreme values (section 4.1.1). The edge dropout provides a significant data augmentation effect with even moderate levels of edge dropout as the network sees a combinatoric scale effect (relative to local node connections). For sparse graphs, this had a minimal impact but was significant for highly connected graphs. A node with 10 connections and 2 of them dropped has 45 different combinations. In reality, the stochastic nature of the dropout process means that drops will not just be $\binom{10}{2}$ in this case, but some combination of the various possibilities (3–4 orders of magnitude seems plausible for this scenario).

The performance drop witnessed for hierarchical data versus the raw suggests another area for research. Our current implementation did not take advantage of the additional edge information available within the hierarchical dataset. This resulted in a small but meaningful reduction in accuracy on MNIST. The additional information introduced by Knyazev et al. (2019) seems particularly useful for large image cases where it would be advantageous to inform the network of the broader region. Our convolutional layer could be modified to incorporate this information.

While we have performed some hyperparameter tuning, there is still a large degree of unexplored space. We believe that even for the nearly saturated MNIST dataset, there are still gains to be acquired (we had at least one approximately 10% reduction in relative error late in testing). We will look to follow up on these questions in future work.

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

## A  APPENDIX

Table 3: MNIST image classification error rates. Models were trained for 100 epochs and results for test error are shown for both raw and Delaunay triangulated MNIST superpixel 75. To avoid biasing to the test set, we selected our model's test values by best training accuracy for a given set of hyperparameters. Our model training also included edge dropout on input with 0.05–0.1 for pruned and an optimized dropout rate for raw (higher depth networks needed more to avoid overfitting). Comparison models are MNIST examples from Fey & Lenssen (2019) showing the performance of their b-spline based method with two pooling strategies. Gray et al. (2020) result is their reported value.

|        | **Ours** | | | **SplineCNN** | | **Gray** |
|--------|------|------|------|-----------|--------------|------|
| Layers | 3    | 4    | 7    | w/ graclus | w/ voxel grid |      |
| Raw    | 4.88 | 2.89 | **0.80** | 2.98  | 2.74         | 0.95 |
| Pruned | 3.29 | 1.87 | **1.04** | 3.66  | 2.88         | -    |

We show error rates for smaller depth networks of our model against MNIST in Table 3 . Results for layer depth 3 and 4 used considerably narrower models than used in the 7 layer version (initial width of 8). At 4 layers deep, our network was considerably faster per epoch than the 2 or 3 layer deep SplineCNN (Fey et al., 2017) or Gray et al. (2020).

After further experiments showed the clear win for deeper networks, we did not go back and attempt to train wide shallow networks. It's quite possible that wider, shallower networks would be able to perform comparably with recent results when properly tuned.

The CIFAR10 superpixel classification results are shown in Table 4. Of note, our method's training accuracy stays considerably closer to test results – which is consistent with our experience on the other datasets. Training accuracies remaining at (or even below) test accuracies was common for our models when sufficient edge dropout was used. The $> 10\%$ difference here is considerably out of the norm when compared to MNIST.

In contrast to MNIST, the hierarchical variant for CIFAR10 offered a considerable performance improvement for our network. Gray et al. (2020) manages to beat our model's accuracy on the raw CIFAR10 superpixel dataset, but we win out on the hierarchical variant. Note that we had not adapted the network to properly use the hierarchical information, so it is particularly surprising to see this result.

For the ModelNet10 results, we used a point cloud representation. Compared to other methods which used the same input data type, our method is competitive, but not exceptional. Minimal hyperparameter tuning was performed so the results should be considered a lower bound on possible performance for this dataset.

Table 4: CIFAR10 image classification results. We were unable to replicate the reported results of Knyazev et al. (2019) using the code provided (they reported 69–73% accuracies depending on the variant). We believe that because we only applied these models naively without special tuning or modifications, they are reasonable results and are included for comparison sake. Also, given the fact that all the models are performing at roughly the same level, it is possible that our recreation of the superpixel dataset suffers from some unknown flaw. The fact that the hierarchical variant shows improved performance for our model (in contrast to other results — section 4), suggests that there is something odd with either the raw or the hierarchical datasets.

| **Superpixel** | Ours | Knyazev et al. (2019) | Gray et al. (2020) |
|---|---|---|---|
| Train Accuracy (%) | 90.02±0.35 | 91.85±0.09 | 99.89±0.01 |
| Test Accuracy (%) | 65.06±0.24 | 60.24±3.18 | **67.65**±67.99 |
| **Hierarchical** | | | |
| Train Accuracy | 86.18±0.41 | 94.28±0.13 | 99.85±0.01 |
| Test Accuracy | **68.94**±0.42 | 63.74±1.10 | 67.96±031 |

Table 5: ModelNet10 shape classification results.

| Model | Reported Accuracy |
|---|---|
| Garcia-Garcia et al. (2016) | 77.6% |
| **Ours** | 86.1% |
| Dominguez et al. (2018) | 93.1% |
| Cheraghian & Petersson (2019) | 94.7% |
| Liu et al. (2018) | **95.3%** |

