# OpenReview forum: "Parameterized Pseudo-Differential Operators for Graph Convolutional Neural Networks"
_ICLR.cc/2021/Conference — Reject_

### Official Review · AnonReviewer4 · 2020-10-28
**Simple operators for unstructured graphs**

**Rating:** 4
**Confidence:** 3

**Review:**

The paper proposes a new graph convolutional layer for graphs with relative position encoding. These types of graphs occur in applications such as meshes, point clouds, and super-pixel neighborhoods. The method extremely simple and generic.

Strengths:
+ The method is simple and can be easily integrated into existing frameworks.
+ The method is more general than approaches designed directly for mesh processing
+ Despite the simplicity, the paper performs well on the Faust node correspondence task. The results are close to that of recent approaches (Haan et al. 2020, Gong et al. 2019) which where directly designed for meshes, while this approach is more general.

Weaknesses:
- Most of the experiments were devoted to super-pixel image classification, MNIST and CIfFAR which isn't a very compelling use case for this type of method. The paper would be stronger with more experiments on datasets like Faust where the data is not grid structured. It would be interesting to see results on point cloud data, where graphs could be constructed using KNN or other methods.
- Limited ablations. The features for message passing are derived from spatial gradients, it would be useful to know which operators are necessary. To my knowledge, without the gradient operators, the network reduces to something more like a generic graph convolutional network. Other parts of the paper say things like "
- The term psuedo differentiable operators isn't defined in the introduction which makes it difficult to understand how this paper relates to other work. It is not clear from to me the novelty compared to (Tencer & Potter, 2020)

Minor Points:
* The paper mentions trying voxel_grid and graclus but does not discuss what theses operations are for
* The organization of the paper could be improved. Experimental details are scattered throughout the paper. Implementation details should be moved from the method sections to the experiments section.

Conclusion
The proposed method is simple and seems to work well on super-pixel and mesh processing tasks. However, limited experiments make it difficult to assess the generality of the method to other tasks

Post-Rebuttal Update: The response from the authors addressed several of my concerns and several clarity issues where fixed in the update paper. However, I don't think the results on ModelNet10 provide strong support for this method. While I don't think it is reasonable to expect this method to outperform other works which are specifically designed for mesh/point cloud inputs (given that this method is more general), I think there needs to be some application outside of super-pixel classification where the proposed method shows an clear advantage.

---

### Official Review · AnonReviewer1 · 2020-10-28
**Are simple differential operators really better?**

**Rating:** 5
**Confidence:** 3

**Review:**

The paper defines simple differential operators at nodes in a graph (gradient, Laplacian) and uses them in the proposed graph convolutional layer. The claim is that simple operators will limit the representation power of the layer leading to better generalization. While this might be true at some level, it goes against the trends in deep learning which move away from such predefined constraints on the representation power of models.

The paper is motivated by inherently non-Euclidean domains but mostly uses super-pixel representations of images for experiments which forces inherently Euclidean data into a non-Euclidean representation. Perhaps not surprisingly, while beating the state of the art in graph convolutional neural nets (CNN) on MNIST/CIFAR, the accuracy is still far from regular CNNs for image classification. I wonder if the choice to use the image data comes from the need to have positional information for the nodes of the graph and how well this requirement is met by other applications where the data is inherently non-Euclidean? The introduction only mentions image data as fitting this requirement. This should be further discussed.

Perhaps a better way to demonstrate the claim that constraining the network to simple representations at each node leads to better generalization would be in the context of limited training data (semi-supervised learning, few shot learning etc). On the other hand, the accuracy of the proposed method is behind the state-of-the-art on the only inherently non-Euclidean dataset used (FAUST) which can be considered limited data (100 examples in total).

---

### Official Review · AnonReviewer3 · 2020-10-29
**Proposed a graph convolutional layer inspired by the pseudo-differential operator**

**Rating:** 5
**Confidence:** 4

**Review:**

This paper proposed a novel graph convolutional layer inspired by the pseudo-differential operator. The proposed conv layer is based on Eqs.(1-3), which is  inspired by the differential operators such as gradient and laplacian operator. The paper claims that this proposed convolutional layer enables to achieve state-of-the-art results on super-pixel image classification.

My major concerns on the work are the limited novelties, insufficient comparisons, and limited significance on the application of super-pixel image classification.

1.  There are several different designs in local convolution over graph nodes in previous works,  such as graph attention convolution, etc. What are the major advantages and novelties of the proposed convolution operators, compared with these previous local convolution designs?

2.  The paper applied the proposed GCN to super-pixel 2D image classification. I have concerns on this application, because the super-pixel-based image recognition using GCN is not a dominant approach in 2D image recognition,  compared with other recognition networks based on regular pixel grids. I suggest that more comparisons on benchmark should be compared, e.g., the 3D shape recognition datasets, or point cloud datasets.

3.  In table I and  II, on the results comparisons on superpixel image recognition, the paper mainly compared with the methods of Knyazev and Gray. I am curious on more comparisons with other GCNs on these datasets if the codes are available.

4. The experimental results show that the proposed network works good on super-pixel image recognition with reduced over-fitting. What are the fundamental reasons that the proposed graph conv. layer is more effective?


---
Post rebuttal comments：
Thanks for the responses. After reading these responses and other reviews, I still has concerns on the justification of proposed convolution compared with other popular graph convolutions, and also the limited experiments on superpixel image recognition.

---

### Decision · Program_Chairs · 2021-01-07
**Final Decision**

**Decision:**

Reject

**Comment:**

All three reviewers expressed consistent concerns on this submission in their reviews. In addition, none of them enthusiastically supported this work during discussion. It is clear this submission does not make the bar of ICLR. Thus a reject is recommended.